# Proton range monitoring using $^{13}$N peak for proton therapy applications

**M. Rafiqul Islam**[1,2], **Mehrdad Shahmohammadi Beni**[3,4], **Chor-yi Ng**[5], **Masayasu Miyake**[3], **Mahabubur Rahman**[6], **Shigeki Ito**[7], **Shinichi Gotoh**[8], **Taiga Yamaya**[9], **Hiroshi Watabe**[1,3]*

**1** Graduate School of Biomedical Engineering, Tohoku University, Sendai, Japan, **2** Institute of Nuclear Medical Physics, AERE, Bangladesh Atomic Energy Commission, Dhaka, Bangladesh, **3** Division of Radiation Protection and Safety control, CYRIC, Tohoku University, Sendai, Japan, **4** Department of Physics, City University of Hong Kong, Kowloon Tong, Hong Kong, **5** Queen Mary Hospital, Pok Fu Lam, Hong Kong, **6** Nuclear Safety Security Safeguard Division, Bangladesh Atomic Energy Regularity Authority, Dhaka, Bangladesh, **7** Mirai Imaging Inc., Iwaki-shi, Japan, **8** GO Proton Japan Inc., Tokyo, Japan, **9** National Institute for Quantum and Radiological Science and Technology, Chiba, Japan

* watabe@cyric.tohoku.ac.jp

**Data Availability Statement:** All data files and the open-source program presented in this work are available from Figshare database (DOI: 10.6084/m9.figshare.16635697.v2).

## Abstract

The Monte Carlo method is employed in this study to simulate the proton irradiation of a water-gel phantom. Positron-emitting radionuclides such as $^{11}$C, $^{15}$O, and $^{13}$N are scored using the Particle and Heavy Ion Transport Code System Monte Carlo code package. Previously, it was reported that as a result of $^{16}$O(p,2p2n)$^{13}$N nuclear reaction, whose threshold energy is relatively low (5.660 MeV), a $^{13}$N peak is formed near the actual Bragg peak. Considering the generated $^{13}$N peak, we obtain offset distance values between the $^{13}$N peak and the actual Bragg peak for various incident proton energies ranging from 45 to 250 MeV, with an energy interval of 5 MeV. The offset distances fluctuate between 1.0 and 2.0 mm. For example, the offset distances between the $^{13}$N peak and the Bragg peak are 2.0, 2.0, and 1.0 mm for incident proton energies of 80, 160, and 240 MeV, respectively. These slight fluctuations for different incident proton energies are due to the relatively stable energy-dependent cross-section data for the $^{16}$O(p,2p2n)$^{13}$N nuclear reaction. Hence, we develop an open-source computer program that performs linear and non-linear interpolations of offset distance data against the incident proton energy, which further reduces the energy interval from 5 to 0.1 MeV. In addition, we perform spectral analysis to reconstruct the $^{13}$N Bragg peak, and the results are consistent with those predicted from Monte Carlo computations. Hence, the results are used to generate three-dimensional scatter plots of the $^{13}$N radionuclide distribution in the modeled phantom. The obtained results and the developed methodologies will facilitate future investigations into proton range monitoring for therapeutic applications.

## Introduction

Considering the currently employed radiation therapy techniques, two types of radiotherapy modes based on photons and protons are used extensively. The ultimate goal of radiation

**Funding:** This research work was funded by Agency for Medical Research and Development (AMED) under Grant Number 20he2202004h0402 and Ministry of Education, Culture, Sports, Science and Technology (MEXT) under Grant Number 21F21103. The funders had no role in study design, data collection and analysis, decision to publish, or preparation of the manuscript.

**Competing interests:** The authors have declared that no competing interests exist.

therapy is to deliver a certain amount of radiation dose to the targeted organs while not affecting healthy organs and cells. In this regard, the use of high-energy proton beams has garnered significant attention worldwide [1] owing to their low lateral scattering, no exit dose, and high dose deposition in the Bragg peak region. Previously, we performed extensive comparisons between photon and proton radiation therapies for pediatric applications [2] and discovered that proton beams can significantly reduce the off-target dose to healthy organs and cells.

However, because of the significant gradient of the dose fall-off primarily after the Bragg peak, the proton therapy technique is sensitive to spatial uncertainties. In other words, the uncertainties in the estimated position of the tumor region can result in excessive dose deposition in non-targeted organs and reduce dose deposition in targeted organs [3,4]. These uncertainties primarily originate from approximations associated with dose calculations, unanticipated anatomical changes, and mispositioning errors during accelerator setup for irradiation. In clinical trials, a setup margin is generally allocated to the target volume to circumvent the effects of these uncertainties.

Several techniques for proton range monitoring have been introduced and discussed. Most of these methods are based on the byproduct of proton beam irradiation on patients. Other typically employed techniques include proton radiography and tomography [5,6], which primarily deliver protons of sufficient energy to the patient to reconstruct planar (two-dimensional, 2D) or tomographic (three-dimensional, 3D) images. In this transmission imaging technique, radiography images are created through the proton's entrance and exit coordinate information provided by a sensitive detector. The primary disadvantage of proton radiography and tomography is the scattering effect, which reduces the resolution of the obtained images [7]. Another direct and cost-effective proton range monitoring technique is the ionoacoustics technique [8,9], which measures acoustic pressure waves for proton range verification. In this technique, the irradiated volume is heated as a result of the deposited radiation dose, and pressure waves are emitted consequently. The acoustic pressure waves are characterized by their amplitude, frequency, and shape, which are governed by the absorbed dose and target material. The ionoacoustics technique offers a direct approach for proton-range verification. However, for the relatively small amplitudes of acoustic signals, this task becomes more challenging. In addition, complexities associated with the coupling between acoustic sensors and human skin exist, rendering this technique laborious. In addition to the abovementioned methods, secondary electron bremsstrahlung can be used for proton range verification [10,11], which uses bremsstrahlung photons generated via charge particle deceleration in matter. Because these photons are of low energy, the method is only applicable to the irradiation monitoring of superficial tumors (i.e., shallow depths). Furthermore, the continuous energy spectrum of bremsstrahlung photons renders it difficult to detect and separate from the background radiation, unlike positron annihilation photons, which have discrete energies. Prompt gamma imaging is another widely employed method for verifying the proton range [12,13]; it uses prompt gamma rays emitted from excited nuclei during the inelastic interactions of incident protons with the target. One significant disadvantage of this approach is its low detector efficiency. Auto-activation positron emission tomography (PET) is another interesting and non-invasive technique that can be used for the range verification of protons; it focuses on measuring photons annihilated from generated positron emitters such as $^{11}$C, $^{15}$O, and $^{13}$N as a result of nuclear interaction between protons and tissues in the body of the patient. The applicability of PET imaging in proton therapy monitoring was previously investigated by several groups [14–20]. In addition, the generated positron emitting radionuclides and their production channel was listed in previous studies; this was reported for proton interaction with human tissues [21].

It is well-known that the $^{16}$O(p,2p2n)$^{13}$N reaction has a relatively low threshold energy (5.660 MeV) [22]. Therefore, by computing the gradient between early and late PET scans, one

can extract the $^{13}$N creation, which is discovered to be associated closely with the Bragg peak. Considering this property and the high sensitivity and spatial resolution of some previously developed PET systems, it would be useful to extensively investigate the underlying mechanism and feasibility of the $^{16}$O(p,2p2n)$^{13}$N nuclear reaction and the generated $^{13}$N peak. The spatial locations of positron emitting radionuclides can be precisely measured using PET or PEM (positron emission mammography) systems around the patient after or during proton irradiation. Furthermore, the correlation between the $^{13}$N peak and the actual Bragg peak should be discussed more comprehensively for different incident proton beam energies. These studies would be useful for proton range monitoring, particularly for therapeutic applications. In the present study, we employed the Monte Carlo (MC) method to simulate the proton irradiation of a homogeneous water-gel phantom. The correlation between the $^{13}$N peak and the actual Bragg peak is discussed in terms of the offset distance for various incident proton energies. The spectral analysis (SA) approach was used to reconstruct the $^{13}$N peak, and the 3D distribution of $^{13}$N radionuclide was obtained. In addition, a standalone open-source computer program, i.e., PeakCalib, was developed to precisely calibrate the $^{13}$N peak with the actual Bragg peak. The obtained results, introduced methodology, and developed computer program would facilitate future developments in the field of proton therapy based on using the $^{13}$N peak for proton range verification. The overall objective of the present work is to investigate the effect of incident proton energy on the production of $^{13}$N positron emitting radionuclides, which in turn can be used to estimate the location of the Bragg-peak. The $^{13}$N and the Bragg-peak found to have an offset distance, and this was computed for wide range of incident proton energy. The current findings, developed tools and introduced approach would lay the pavement for future investigations and advancement in the field of proton range verification.

## Materials and methods

### MC method

In the present study, we used the Particle and Heavy Ion Transport Code System (PHITS) code version 3.25 [23]. It is a general-purpose MC simulation code that uses the Jet AA microscopic transport model (JAM) [24] and JAERI quantum molecular dynamics (JQMD) [25] to describe intermediate and high-energy nuclear reactions. Both the JQMD and JAM physical models can be used to describe the dynamic stages of the reactions. In the present study, a water-gel phantom measuring 10 cm × 10 cm × 40 cm was modeled. A schematic illustration of the modeled geometry is shown in Fig 1. The material composition of the modeled water-

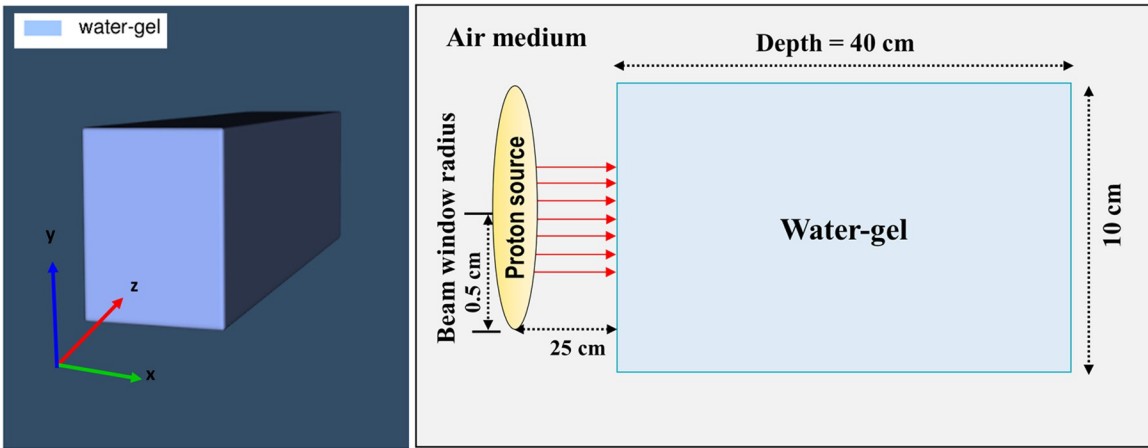

**Fig 1. Schematic diagram of water-gel phantom in three dimensions.**

**Table 1. Density and material composition (fraction by weight for each nuclei) used in present model.**

| Material | Density (g/cm$^3$) | $^1$H (%) | $^{12}$C (%) | $^{16}$O (%) |
|----------|---------------------|-----------|--------------|--------------|
| Water-gel | 1.010 | 11.00 | 4.650 | 84.35 |

gel is presented in Table 1. The composition of water-gel phantom that was reported in previous investigations have rather very low nitrogen content [26,27]. Generally, some water-gel phantoms were produced experimentally by mixing agar powder ($C_{14}H_{24}O_9$) with water ($H_2O$), with the ratio of 1/100 (i.e., agar powder/water). Therefore, we have not considered nitrogen ($^{14}$N) in the modelled water-gel. The modeled incident proton beams considered were monoenergetic pencil-like beams of energies 80, 160, and 240 MeV with protons measuring 1 cm in diameter emitted along the positive z-axis. The location where the incident proton beam is irradiated is additionally shown in Fig 1. In the modelled irradiation setup, 25 cm of air gap was considered between the proton beam and the phantom. To reduce statistical uncertainties associated with the MC method, we launched $10^9$ protons from the modeled beam. The Monte Carlo method is well-established in simulation of radiation transport. The stochasticity of interaction of radiation with matter can be conveniently considered using the Monte Carlo method; this is mainly accomplished by using pseudo random numbers at which determines the interaction with different nuclei and sampling the angular and energy distribution. Considering such stochasticity, the statistical analysis of the results would be important, in a way that low relative error in the estimated results would be desired. More details regarding the MC simulation and modeling are available in our previous publications and the references therein [28–31].

Upon the interaction of the protons with the target elements, different positron emitters were produced, which was primarily due to inelastic nuclear interactions. The modeled water-gel was primarily composed of oxygen (see Table 1). In addition, it is noteworthy that hydrogen does not produce stable positron emitters; therefore, it was not considered in the present discussion. In present work, we have employed a homogeneous water-gel phantom mainly to eliminate any complex geometrical effect that might arise from the heterogeneities, such as those can be found in human tissue; this is to investigate the correlation between the production of $^{13}$N radionuclides and different incident proton energies. A list of primary nuclear reactions and positron emitters generated as a result of this particular reaction is summarized in Table 2. The created isotope, half-life, reaction channel, and threshold reaction energy are listed in Table 2, these data were taken from Ref. [21].

The absorbed dose vs. depth and the spatial distribution of the positron-emitting radionuclides (i.e., $^{11}$C, $^{15}$O, and $^{13}$N) were compared along the z-axis of the incident proton beam. The obtained results were normalized to the primary incident proton (see Ref. [2]). The tally results obtained from Monte Carlo simulations are mostly normalized per primary source particle by the Monte Carlo simulation package, as the absolute values would have no physical meaning. Therefore, the obtained results were normalized to the primary incident proton.

**Table 2. Created isotopes, half-life, reaction, and threshold reaction energy.**

| Isotope | Half-life (min) | Reaction | Threshold (MeV) |
|---------|-----------------|----------|------------------|
| $^{11}$C | 20.39 | $^{12}$C(p,pn)$^{11}$C | 20.61 |
| $^{15}$O | 2.037 | $^{16}$O(p,pn)$^{15}$O | 16.79 |
| $^{13}$N | 9.965 | $^{16}$O(p,2p2n)$^{13}$N[a] | 5.660 |

(a): (p,2p2n) is inclusive of (p,α).

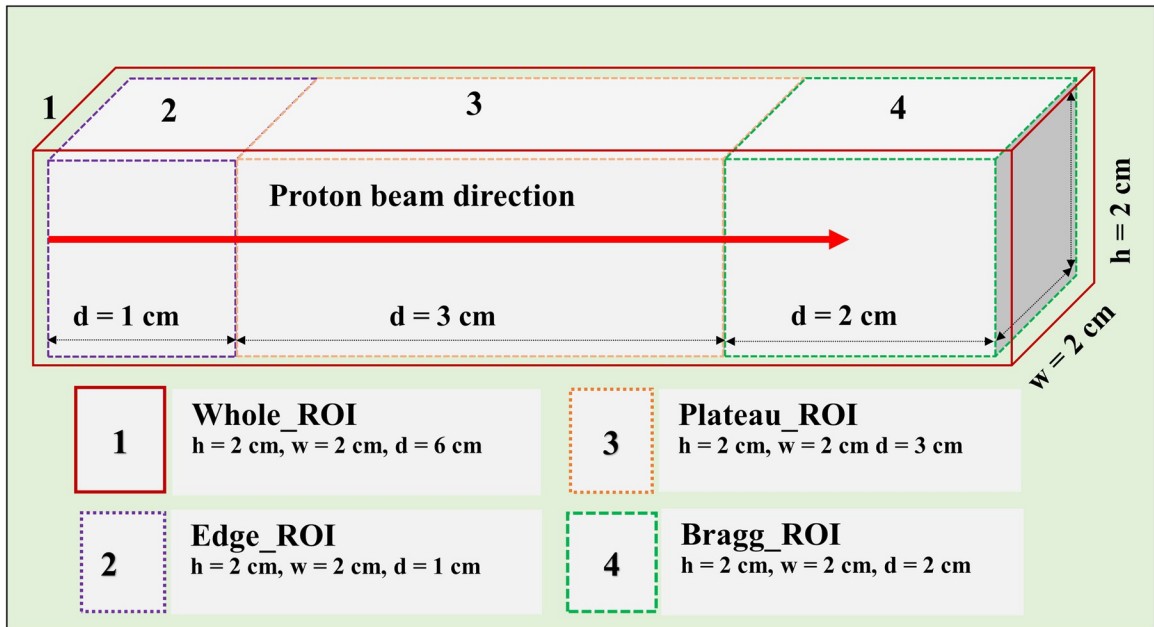

**Fig 2. Side view of four regions of interests (ROIs) with their respective height (*h*), width (*w*), and depth (*d*) values.**

Subsequently, the obtained data were converted into activity by considering the physical half-life of each positron-emitting radionuclide. The constructed dynamic frames were generated at 1 min intervals until 75 min. Our previously developed PyBLD software [32] (link: http://www.rim.cyric.tohoku.ac.jp/software/pybld/pybld.html) was used to analyze the output data from the PHITS. The obtained images were analyzed using the software, A Medical Image Data Examiner (version 1.0.4) (link: http://amide.sourceforge.net) [33]. The one-dimensional (1D) depth dose and activity profiles were obtained, which provided information regarding the location of the activity and its distribution. The 3D data were analyzed by considering four different regions of interest (ROIs): (1) whole, (2) edge, (3) plateau, and (4) Bragg-peak region. These four regions, with their respective heights, widths, and depths, are shown in Fig 2.

Subsequently, the results from these four regions were used in the spectral analysis calculations for the data obtained 60 min after irradiation. In addition, 1D time profiles from the whole region data for [11]C, [15]O, and [13]N radionuclides were calculated for 15, 20, 30, 60, and 75 min after irradiation. The presence of proton-induced radionuclides was confirmed from the obtained results. Finally, 3D scatter plots were generated by performing spectral analysis on each voxel. Additionally, the 3D distribution of [13]N was obtained and visualized.

## SA approach

SA is widely performed to identify kinetic components (i.e., tracers) in each voxel of a PET image in the field of nuclear medicine [34]. SA does not require non-linear optimization for compartmental modeling. More details regarding compartmental modeling are available in our previous publications and our recently developed compartmental software [35–37]. Compartmental modelling refers to the of modelling substance transport in a system consisting of multiple compartments (i.e., distinct regions/voxels), which is characterized by the transfer rates among the relevant compartments. The variations of certain substances or, more generally, the radionuclides in different compartments could be explained using sets of differential equations. SA requires relatively low computational resources; nonetheless, it can yield voxel-

by-voxel functional images. Each voxel of the PET image contains several positron-emitting radionuclides because of the interaction between the incident proton beam and the target elements. In this study, we performed SA to distinguish the $^{13}$N component from other positron-emitting radionuclides in each voxel. The counts as a function of time ($t$) in the voxel ($v$) of the PET image, denoted as $C_v(t)$, as a function of the incident proton beam profile of A($t$), can be expressed as

$$C_v(t) = \sum_{j=1}^{M} A(t) \otimes \alpha_j e^{-\beta_j t}, \tag{1}$$

where $\otimes$ is the convolution operation; $M$ represents different types of radionuclides produced, numbered from $j = 1$ to $j = M$; $\alpha_j$ and $\beta_j$ are the initial radioactivity and decay constant of radionuclide $j$, respectively. In this study, we assumed an impulse function for A($t$). Datasets for $\beta_j$ were first prepared (by default, the range of $\beta$ is from $10^{-4}$ to $0.1$ s$^{-1}$ and is logarithmically divided, with $M = 1000$), and each A($t$) $\otimes$ exp ($-\beta_j t$) (impulse response function) was pre-calculated. Subsequently, sets of $\alpha_j$ were solved using a non-negative least squares estimator, which was used to solve Eq (2).

$$C_v(t) = \sum_{j=1}^{M} \alpha_j IRF_j \tag{2}$$

The estimated sets of $\alpha_j$ were linear; hence, SA can determine groups of $\alpha_j$ without requiring any iterations and therefore promptly calculate the sets of $\alpha$ and $\beta$ in each voxel. Ideally, we wish to obtain a few positive sets of $\beta$ that correspond to the decay constants of the produced radionuclides. However, practically, several peaks of $\beta$ will appear owing to numerical errors arising from the discreteness of $\beta$. Therefore, we computed the numerical value $S_v = \Sigma_{j=1} \alpha_j \beta_j$ for each voxel ($v$). $S_v$ enhances the production of short half-life radionuclides (e.g., $^{13}$N with a large $\beta$) and suppresses that of longer half-life radionuclides (e.g., $^{11}$C with a small $\beta$). The threshold value of $\alpha\beta$ was set to > 1.5, which removes the background region.

In realistic measurements using PET system, the measured signal could be weak and therefore it generates noisy images. There are various ways to circumvent the issue with weak signal and in turn denoise PET images. Recently, Guo *et al.* [38] introduced a novel kernel graph filtering method that could significantly tackle the issue with noisy PET images as a result of weak signal. The study performed by Guo *et al.* [38] was tested extensively using simulated and real life in-vivo dynamic PET datasets. The authors showed that the proposed method significantly outperforms the existing methods in sinogram denoising and image enhancement of dynamic PET at all count levels, and especially at low counts which measured signal from isotopes are weak. Therefore, the issue with weak signals that may create difficulties in realistic measurements could be solved rather effectively using denoising methods. In addition, the total body PET scanner is another system that can be used to solve the issue with weak signals; this scanner has 200 cm axial field of view (FOV) and 40 times higher sensitivity than conventional PET systems [39].

## Results and discussion

### MC computations

The 1D profile of dose vs. depth was obtained for energies of 80, 160, and 240 MeV. Similarly, the relative distributions of positron-emitting radionuclides (i.e., $^{11}$C, $^{15}$O, and $^{13}$N) were calculated along the z-axis (i.e., along the incident proton beam). The obtained results are shown in Fig 3, where the sum of the activities of all three radionuclides are shown in the same plot.

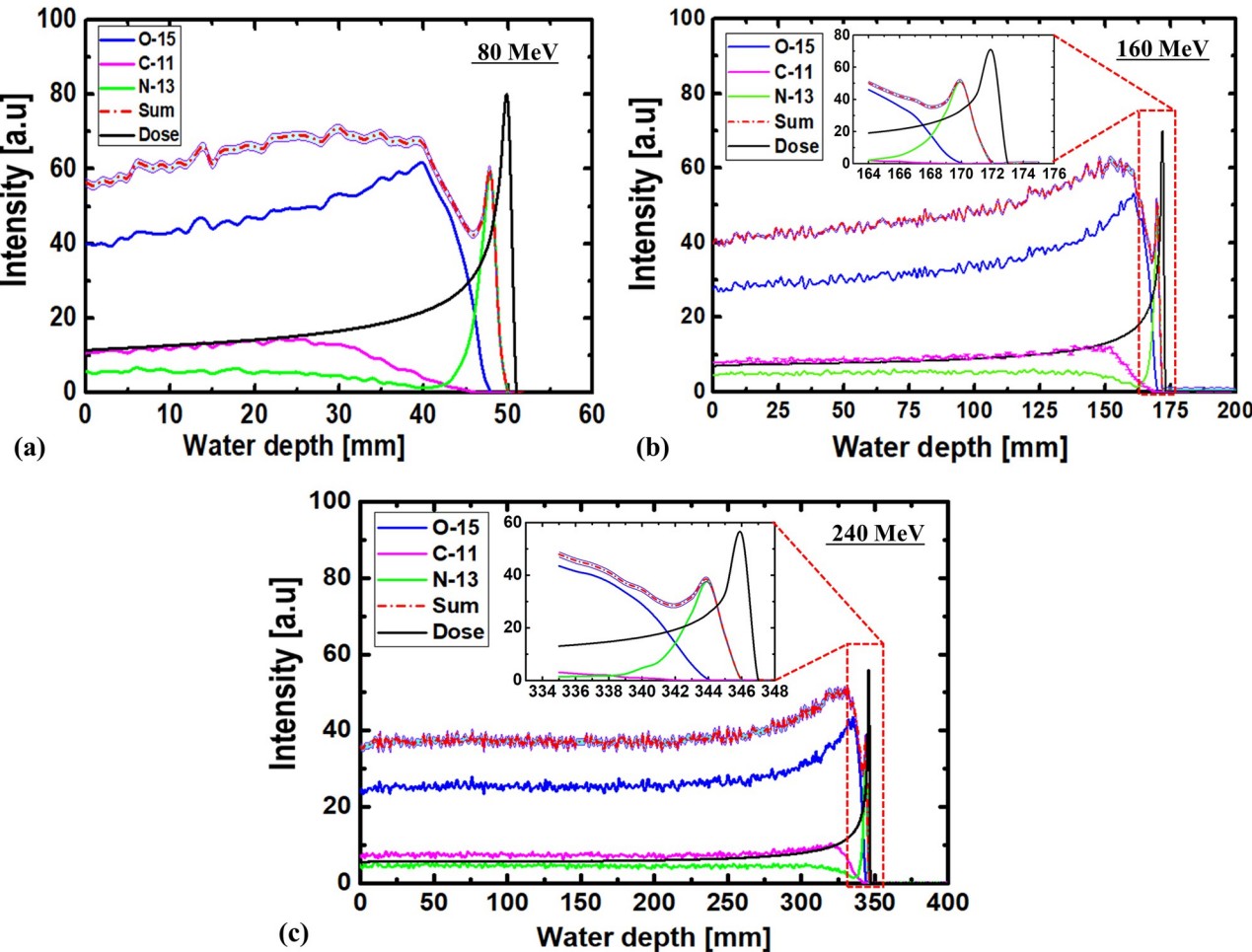

**Fig 3.** 1D dose and relative distribution of positron-emitting radionuclides obtained along incident beam direction, immediately after proton irradiation (i.e., t = 0) for (a) 80, (b) 160, and (c) 240 MeV incident proton energies. Statistical uncertainties associated with Monte Carlo computation is shown for sum curve.

The average relative errors were 0.169, 0.072, and 0.147 for energies 80, 160, and 240 MeV, respectively. The sum represents the combination of radionuclides possessing radio activities of $^{11}$C ($T_{1/2} \approx 20$ min), $^{15}$O ($T_{1/2} \approx 2$ min), and $^{13}$N ($T_{1/2} \approx 10$ min), immediately (at time t = 0) after proton irradiation. The dose shown in Fig 3 was scaled such that it can be plotted in the same graph as the radionuclide activities. As shown in Fig 3, the production of $^{11}$C and $^{15}$O decreased before the Bragg peak and in the fallout region. However, the production of $^{13}$N generated a peak near the Bragg peak region. The primary reason causing the earlier decline of $^{11}$C and $^{15}$O as compared with $^{13}$N was that the threshold energy for the production of $^{11}$C (20.61 MeV) and $^{15}$O (16.79 MeV) was higher than that for $^{13}$N (5.660 MeV). It is noteworthy that upon the interaction of protons with matter, the proton will lose energy; therefore, lower-energy protons are to be expected in the deeper region of the water-gel phantom. At superficial depths, more high-energy protons will be present; therefore, the required threshold energy for $^{11}$C and $^{15}$O production will be satisfied. However, as the depth increases and the proton energy decreases, the dominant production reaction will be $^{16}$O(p,2p2n)$^{13}$N, which has a relatively lower threshold energy. Hence, the byproduct of this reaction (i.e., $^{13}$N) is expected to be closer to the Bragg peak region. Considering the Bragg peak and the peak at which the $^{13}$N

**Table 3. Proton range comparison for three different incident energies in water-gel from PHITS and SRIM [40].**

| Energy (MeV) | Range from PHITS | Range from SIRM | Absolute deviation |
|---|---|---|---|
| 80 | 59 mm | 58 mm | 1 mm |
| 160 | 199 mm | 197 mm | 2 mm |
| 240 | 389 mm | 395 mm | 6 mm |

radionuclide was created, the distance offset was discovered to be 2.0, 1.9, and 2.0 mm for 80, 160, and 240 MeV, respectively. In other words, the depths at which the Bragg-peak and the peak from $^{13}$N were observed were 49.8 and 47.8 for 80 MeV, 171.8 and 169.9 mm for 160 MeV, and 345.9 and 343.9 mm for 240 MeV. The deviation or the distance offset between the Bragg peak and the 13N peak was likely due to the threshold energy for the $^{16}$O(p,2p2n)$^{13}$N reaction. Based on the definition of the Bragg peak, it is clear that the dose reaches its maximum value at a depth near the end of the particle range, which implies that the incident particle energy will reach its minimum and be lower than 5.660 MeV (i.e., $^{13}$N produces the reaction threshold energy). Therefore, the peak from $^{13}$N and the actual Bragg peak would be located at different depth positions in the water-gel phantom. However, our calculations show that the offset distance was insignificant. This is similarly indicated in Fig 3(a)-3(c) for incident proton energies of 80, 160, and 240 MeV, respectively. For reference, the range of protons for three different incident energies in the water-gel based on the PHITS and SRIM is shown in Table 3 [40] (link: http://www.srim.org/). The PHITS Monte Carlo package computes the average stopping power for the charged particles and nuclei either using the ATIMA package [41].

The computations were performed using three different energies of 80, 160, and 240 MeV emitting along the positive z-axis from a circular source with a diameter of 1 cm. The source was used to irradiate the water-gel phantom, and the results were obtained using the PHITS MC package. Table 3 shows a comparison of the estimated proton range in the water-gel phantom based on our computations using the PHITS and the standard and widely used SRIM. The deviation between the estimated ranges is shown in Table 3. The proton ranges in the water-gel phantom estimated from the PHITS and SRIM showed good agreement. The estimated proton range between the PHITS and SRIM differed slightly owing to the different models and tabulated data used to explain proton straggling and interaction with matter. However, the deviation was relatively small compared with the overall average range for each incident proton energy. For example, considering the 240 MeV incident beam energy, the overall average range based on the PHITS and SRIM was 392 mm, and the deviation was only 1.53% of the overall average range—this can be considered negligible. In addition, the comparison between the estimated range values serves as a good benchmark for our developed MC model.

Because an offset was present between the generated $^{13}$N peak and the actual Bragg peak, a wider incident proton energy range should be considered to precisely verify the distance offset. Therefore, we used our developed model to investigate the distance offset for incident proton energies ranging from 45 to 250 MeV with an interval of 5 MeV. This energy range encompassed the most widely used proton energies used in therapeutic applications. For simplicity, they were obtained immediately after proton irradiation ($t = 0$). The obtained numerical results for the actual Bragg peak, $^{13}$N peak location, and distance offset (Bragg-peak location–$^{13}$N peak location) are shown in Table 4.

Based on the obtained results shown in Table 4 for various incident proton energies, it is clear that the offset distance ranged from 1.0 to 2.0 mm, which is within the acceptable range. The obtained data can be used for the future calibration of the measured $^{13}$N peak to the actual

**Table 4. Comparison between actual Bragg peak and ¹³N peak location in water-gel phantom with offset distance (Bragg-peak location– ¹³N peak location).**

| Energy (MeV) | Bragg peak (mm) | ¹³N peak (mm) | Offset (mm) | Energy (MeV) | Bragg peak (mm) | ¹³N peak (mm) | Offset (mm) |
|---|---|---|---|---|---|---|---|
| 45 | 17.0 | 15.0 | 2.0 | 150 | 153.0 | 152.0 | 1.0 |
| 50 | 21.0 | 19.0 | 2.0 | 155 | 163.0 | 161.0 | 2.0 |
| 55 | 25.0 | 23.0 | 2.0 | 160 | 172.0 | 170.0 | 2.0 |
| 60 | 29.5 | 28.0 | 1.5 | 165 | 181.0 | 180.0 | 1.0 |
| 65 | 34.0 | 32.0 | 2.0 | 170 | 191.0 | 189.0 | 2.0 |
| 70 | 39.0 | 37.0 | 2.0 | 175 | 201.0 | 199.0 | 2.0 |
| 75 | 44.0 | 42.0 | 2.0 | 180 | 211.0 | 209.0 | 2.0 |
| 80 | 50.0 | 48.0 | 2.0 | 185 | 221.0 | 219.0 | 2.0 |
| 85 | 56.0 | 54.0 | 2.0 | 190 | 232.0 | 230.0 | 2.0 |
| 90 | 62.0 | 60.0 | 2.0 | 195 | 242.0 | 241.0 | 1.0 |
| 95 | 68.0 | 66.0 | 2.0 | 200 | 253.0 | 251.0 | 2.0 |
| 100 | 75.0 | 73.0 | 2.0 | 205 | 264.0 | 262.0 | 2.0 |
| 105 | 81.5 | 80.0 | 1.5 | 210 | 275.0 | 274.0 | 1.0 |
| 110 | 88.5 | 87.0 | 1.5 | 215 | 287.0 | 285.0 | 2.0 |
| 115 | 96.0 | 94.0 | 2.0 | 220 | 298.0 | 296.0 | 2.0 |
| 120 | 103.5 | 102.0 | 1.5 | 225 | 310.0 | 308.0 | 2.0 |
| 125 | 110.5 | 109.0 | 1.5 | 230 | 322.0 | 320.0 | 2.0 |
| 130 | 119.0 | 117.0 | 2.0 | 235 | 334.0 | 332.0 | 2.0 |
| 135 | 127.0 | 126.0 | 1.0 | 240 | 345.0 | 344.0 | 1.0 |
| 140 | 136.0 | 134.0 | 2.0 | 245 | 358.0 | 356.0 | 2.0 |
| 145 | 145.0 | 143.0 | 2.0 | 250 | 371.0 | 369.0 | 2.0 |

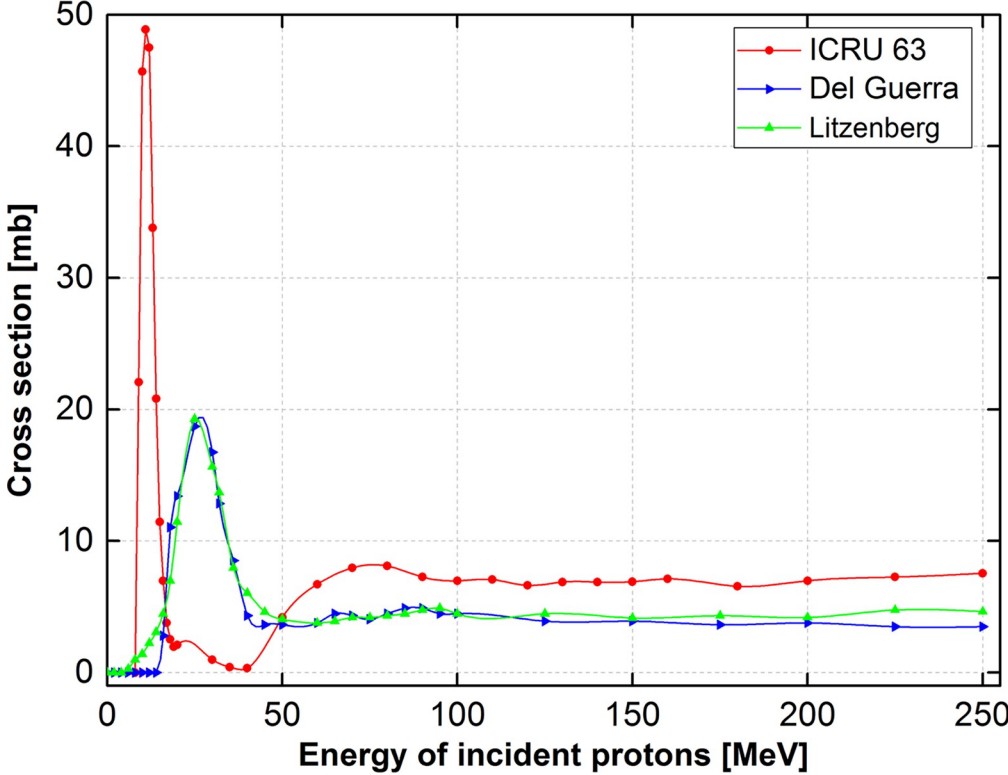

**Fig 4. Energy-dependent cross-section data for ¹⁶O(p,2p2n)¹³N nuclear reaction reported by ICRU 63 [42], Del Guerra [16], and Litzenberg [20], these were taken from Ref. [21].**

Bragg-peak location. It was observed that the distance offset values did not fluctuate significantly for different incident proton energies. This is primarily due to the approximately flat energy-dependent cross-section data for the $^{16}$O(p,2p2n)$^{13}$N reaction in the incident proton energy range of 37.5–250 MeV. The energy-dependent cross-section data for the $^{16}$O (p,2p2n)$^{13}$N nuclear reaction reported by (1) ICRU 63 [42], (2) Del Guerra [16], and (3) Litzenberg [20], these were taken from Ref. [1] and are shown in Fig 4.

The current computations were performed at intervals of 5 MeV. Considering the intermediate energies that might be used in certain irradiation facilities, we developed a standalone open-source peak calibration computer program, i.e., PeakCalib, which reports offset distance values with an energy interval of 0.1 MeV using linear and non-linear spline interpolation techniques. Details regarding the PeakCalib program and the obtained results are provided in Appendix A. The PeakCalib program is distributed and the program can be freely downloaded (from a free public repository), recompiled, and redistributed without any restrictions.

The 2D time-dependent images and their respective intensity profiles are shown in Figs 5–7 for 80, 160, and 240 MeV, respectively. They were predicted from the radioactive decay curve using the PHITS MC computer program. The obtained results were for time ranges from 15 to 75 min, which translates to a 60 min dataset. A scaling factor similar to that used to obtain the

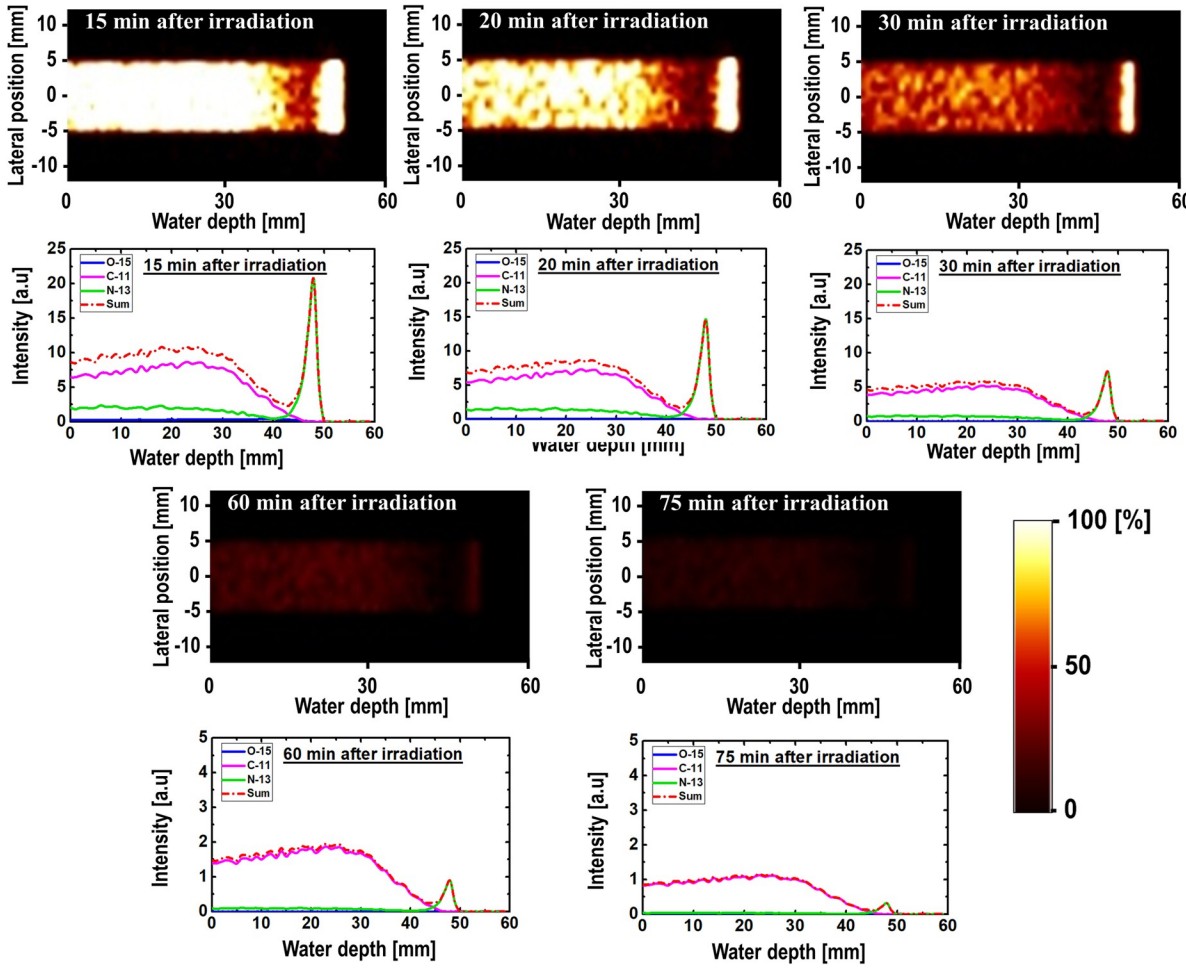

**Fig 5. 2D images and time-dependent activity of three positron-emitting nuclei for 80 MeV incident proton energy.**

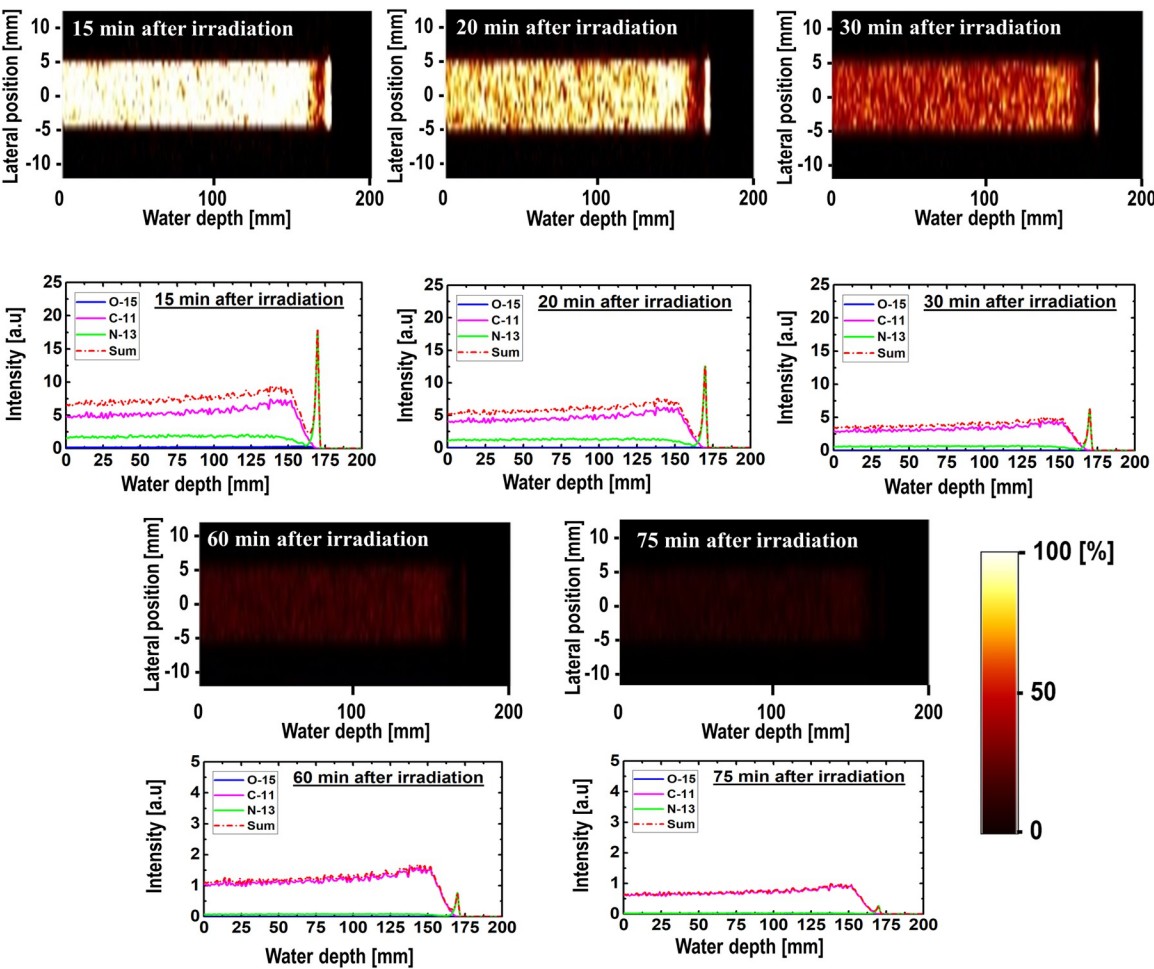

**Fig 6. 2D images and time-dependent activity of three positron-emitting nuclei for 160 MeV incident proton energy.**

results shown in Fig 3 was used in this case, and the y-axis of the 1D profiles was labeled as the relative intensity. The relatively short-lived $^{15}$O ($T_{1/2}$ = ~2.0 min) nuclei spectrum vanished almost completely after 30 min in the time-dependent profile data. The primary observable peak that was similar to the Bragg peak originated from the $^{13}$N nuclei. This trend was observed for all three simulated incident proton beam energies (i.e., 80, 160, and 240 MeV). In fact, the peak from the $^{13}$N nuclei was present for most of the simulated time values. However, owing to its relatively short half-life, the $^{13}$N peak disappeared for longer time values, such as 75 min. It is arguable that such long time durations (e.g., 75 min) will not benefit therapeutic applications; however, it is interesting to observe the presence of a $^{13}$N peak up to 60 min intervals and the dominance by the long-lived $^{11}$C radionuclides for long time durations. The creation of a $^{13}$N peak or any other positron-emitting radionuclides are affected primarily by two factors: (1) their production rate and (2) their decay rate. The two main controlling parameters are the incident proton energy and the half-life for the production and decay of these radionuclides. For example, $^{13}$N has a shorter half-life than $^{11}$C; however, it has a lower threshold energy for its creation compared with $^{11}$C. It is important to account for these two factors simultaneously when analyzing the results. Considering these two factors, for relatively long durations, only the $^{11}$C spectrum can be observed owing to its relatively long half-life. In fact,

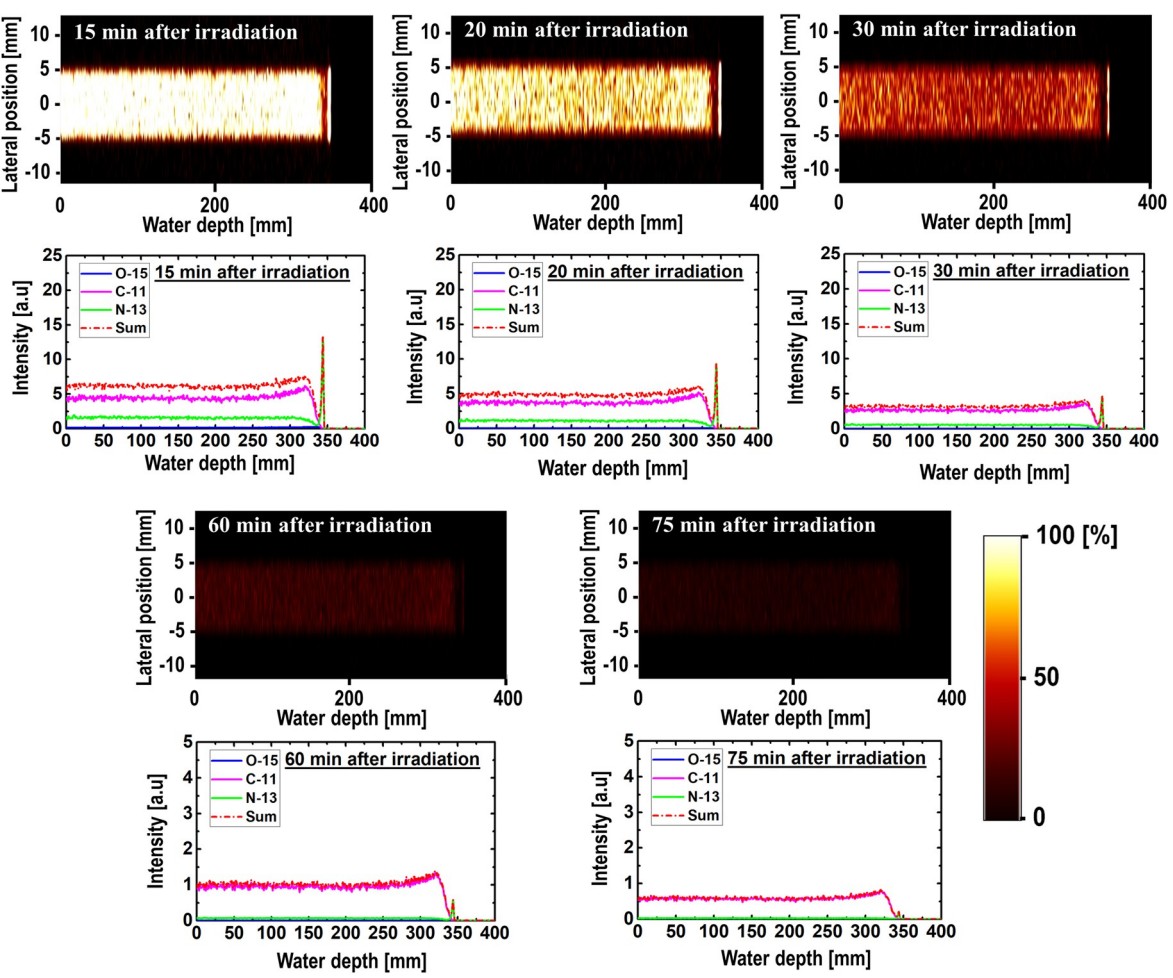

**Fig 7. 2D images and time-dependent activity of three positron-emitting nuclei for 240 MeV incident proton energy.**

the $^{11}$C spectrum dominates regions in the phantom with proton energies equal to or higher than its nuclear reaction threshold energy. Comparing the results of different incident proton beam energies, we observed a distinct $^{13}$N peak in the time range of interest in medical imaging. Regarding the clinical significance of our study, it needs to be noted that intensity of the positron emitting radionuclides can be measured as long as they are being produced (i.e., when annihilation photon is emitted from the patient's body). Furthermore, measuring annihilation photon provides mobility of the patient. The patient does not have to stay on the treatment couch for the measurements. The measurements can be performed in another place. In addition, by measuring annihilation photons using PET system, relative time trend would be measured rather than absolute photon counts. Therefore, it would not be necessary to start the measurements immediately after proton irradiation. Having such flexibility would in fact be beneficial in realistic clinical treatment conditions.

## SA

SA was performed on the dynamic time-dependent activity results. The results shown in Fig 8 (a) are those of 2D images with their respective 1D profiles obtained via SA. SA was performed for three different incident proton energies of 80, 160, and 240 MeV. The peak positions of

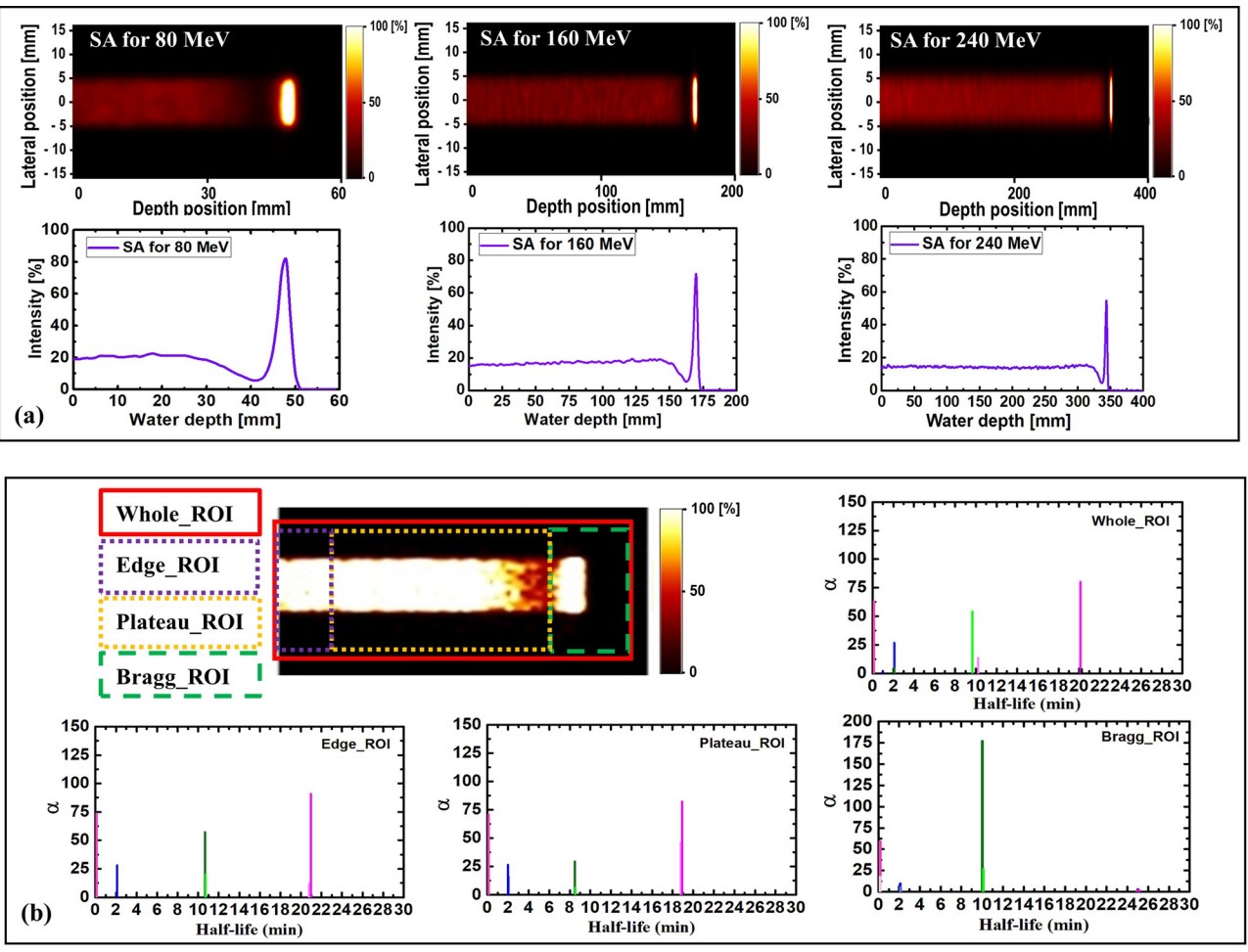

**Fig 8.** (a) SA images for 80, 160, and 240 MeV, respectively; (b) spectral analysis for different ROIs having 80 MeV incident proton beam energy.

SA-extracted $^{13}$N were consistent with the simulated $^{13}$N peak positions for the three energies presented in Figs 9–11. The production of the $^{13}$N peak via SA indicates a promising application of SA for separating the $^{13}$N peak from other positron-emitting radionuclides; this approach would be useful for analyzing the experimentally obtained data.

The selected ROIs at which SA was performed are shown in Fig 2. The ROIs contained whole, edge, plateau, and peak regions (see Fig 2), and the results obtained via SA for the 80 MeV incident proton beam energy are shown in Fig 8(b). For all ROIs, the x-axis represents the half-life (i.e., log(2)/β) for the extracted radionuclides, and the y-axis represents the concentration of radionuclides, labeled as α. Based on the SA results for the whole region shown in Fig 8, it was clear that the contributions from $^{11}$C and $^{13}$N were similar. However, the contribution from the $^{15}$O radionuclides was approximately one-half those of $^{11}$C and $^{13}$N. The SA results of the edge ROI primarily comprised those of relatively long-lived radionuclides; in other words, the contribution from the long-lived radionuclides (e.g., $^{11}$C) was greater than those of $^{15}$O and $^{13}$N. Considering the plateau ROI, it was discovered that $^{11}$C offered the greatest contribution, whereas $^{15}$O and $^{13}$N indicated similar levels of contribution. Finally, the Bragg-peak ROI indicated the greatest contribution from the $^{13}$N radionuclides, whereas the contributions from $^{11}$C and $^{15}$O were negligible.

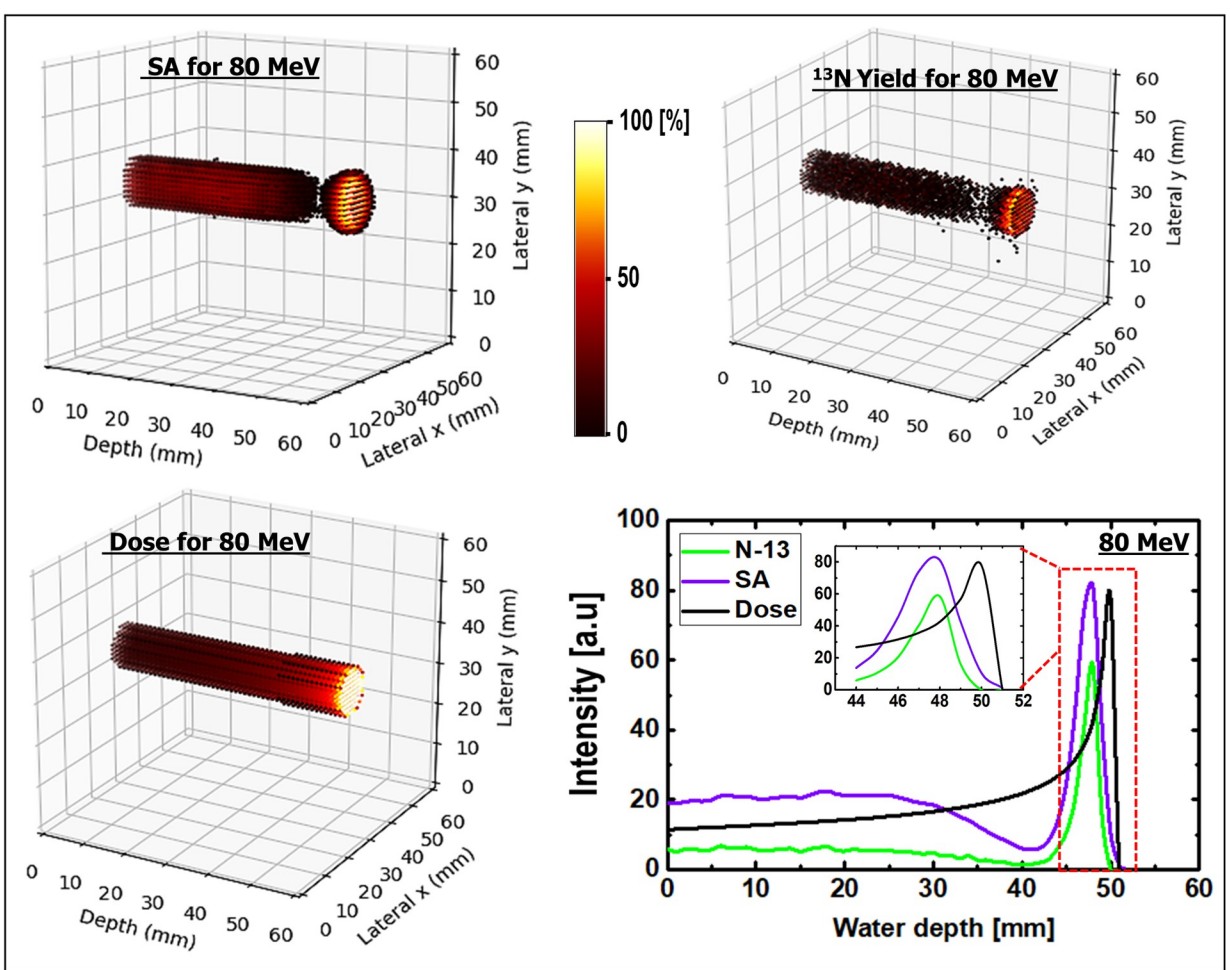

**Fig 9. 3D scatter visualization and 1D profiles of SA image; $^{13}$N production and dose for 80 MeV incident proton energy.**

## 3D visualizations

For a better visualization of the Bragg peak and the peak from $^{13}$N, a 3D SA image was generated. The $^{13}$N production, dose, generated 3D SA image, and 1D profiles are shown in Figs 9–11 for incident proton beam energies of 80, 160, and 240 MeV, respectively. The 3D plots from the SA for incident proton energies of 80, 160, and 240 MeV showed a distinct creation of the Bragg peak; this is another promising approach for verifying the results, particularly those obtained experimentally. Similarly, the 3D plots for the $^{13}$N yield indicated the creation of a peak near the end of the range of the primary particles. Comparing the abovementioned two plots based on a 1D profile, it was observed that the $^{13}$N peak was created near the Bragg peak for all three different incident proton energies. In addition, the 3D dose distributions for the three different incident beam energies indicate that the dose value increased with depth in the water-gel phantom. 3D visualizations would benefit the investigation of inhomogeneous organs (those with irregular geometries) such as the lungs, head, and neck. In fact, the calculation of dose distribution for treatment planning and proton beam positioning are more complex for inhomogeneous organs. Based on the 1D profiles shown in Figs 9–11 for incident proton beams of energies 80, 160, and 240 MeV, respectively, it was observed that the prediction of the $^{13}$N peak based on SA was consistent with the computed $^{13}$N peak from MC

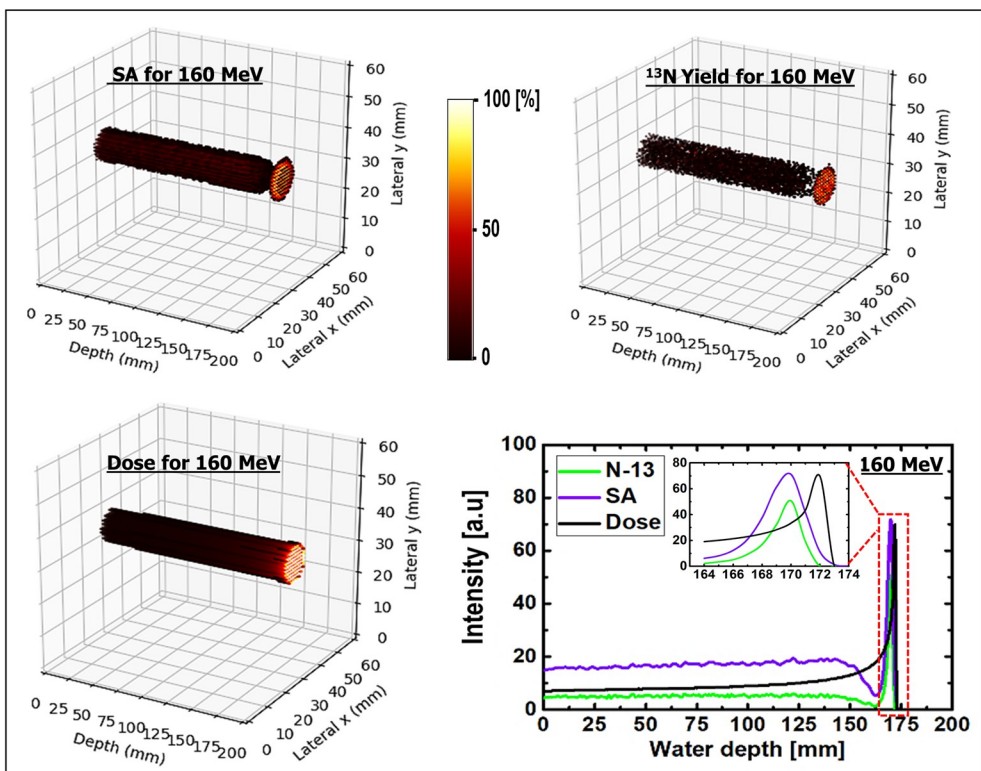

**Fig 10. 3D scatter visualization and 1D profiles of SA image; $^{13}$N production and dose for 160 MeV incident proton energy.**

computations (see Table 4 for the numerical values). Therefore, SA is effective for analyzing experimental data obtained from PET systems.

## Conclusions

Herein, the concept of proton range monitoring using the $^{13}$N peak was discussed for various incident proton energies. The MC method using the PHITS package was used to obtain the production of positron-emitting radionuclides, namely $^{11}$C, $^{15}$O, and $^{13}$N, in the simulated water-gel phantom. Subsequently, the generated $^{13}$N peak was compared with the actual Bragg peak for various incident proton energies, i.e., those from 45–250 MeV, which is within the range of interest for therapeutic applications. The offset distance between the $^{13}$N peak and the actual Bragg peak was primarily due to the threshold energy of the $^{16}$O(p,2p2n)$^{13}$N nuclear reaction. The fluctuations in the offset distance, which were relatively mild for the energy range investigated, were correlated with the energy-dependent cross-section data for the $^{16}$O (p,2p2n)$^{13}$N nuclear reaction. In addition, we developed an open-source computer program to perform linear and non-linear cubic spline interpolation; the program can obtain the offset distance with an energy interval of 0.1 MeV. In addition, SA was performed to analyze the results, which indicated significant $^{13}$N production when compared with other radionuclides ($^{11}$C and $^{15}$O) in the Bragg ROI. SA will benefit future experimental studies as it can separate the $^{13}$N peak from other positron-emitting radionuclides for proton range monitoring. Additionally, the obtained results and the tools developed in the present study will benefit future investigations. In future works, we aim to investigate the production of 13N and other positron emitting radionuclide by irradiating heterogenous phantoms with monoenergetic and spread-out Bragg-peak (SOBP) proton beams.

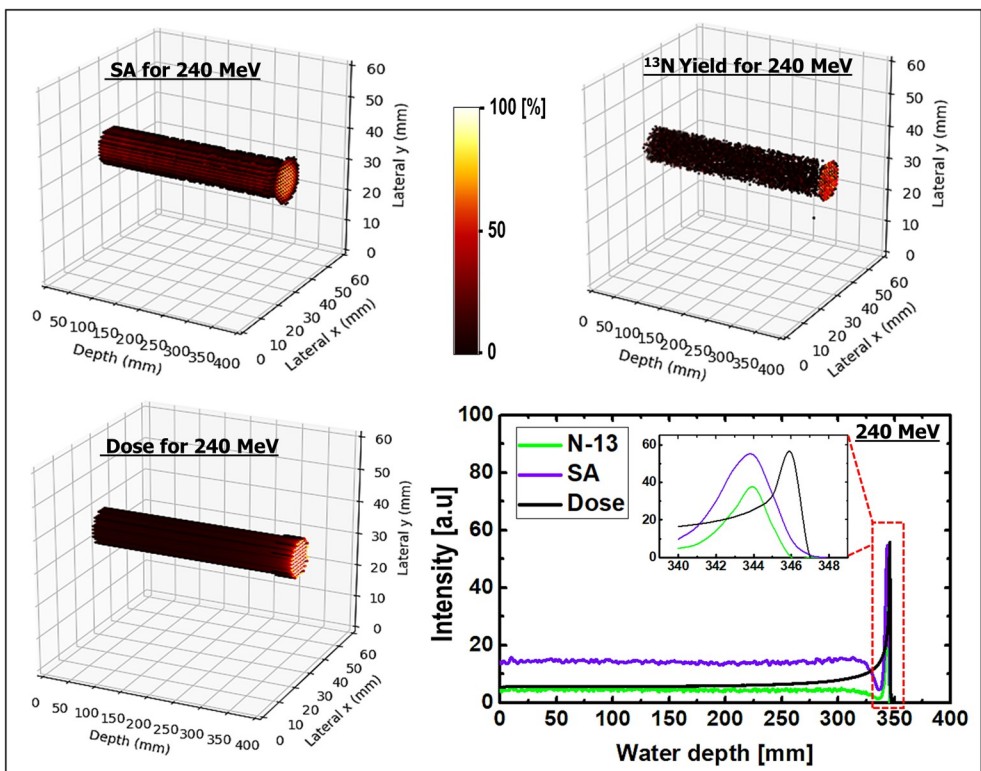

**Fig 11. 3D scatter visualization and 1D profiles of SA image; [13]N production and dose for 240 MeV incident proton energy.**

## Supporting information

**S1 Appendix.**
(DOCX)

## Acknowledgments

The authors thank the Cyclotron and Radioisotope Center (CYRIC) for their technical support.

## Author Contributions

**Conceptualization:** M. Rafiqul Islam, Hiroshi Watabe.

**Data curation:** M. Rafiqul Islam, Mehrdad Shahmohammadi Beni, Masayasu Miyake, Shigeki Ito, Shinichi Gotoh.

**Formal analysis:** M. Rafiqul Islam, Mehrdad Shahmohammadi Beni, Chor-yi Ng.

**Funding acquisition:** Hiroshi Watabe.

**Investigation:** M. Rafiqul Islam, Mehrdad Shahmohammadi Beni, Chor-yi Ng.

**Methodology:** Mehrdad Shahmohammadi Beni.

**Project administration:** Hiroshi Watabe.

**Resources:** Hiroshi Watabe.

**Software:** Mehrdad Shahmohammadi Beni, Hiroshi Watabe.

**Supervision:** Hiroshi Watabe.

**Validation:** M. Rafiqul Islam, Mehrdad Shahmohammadi Beni, Chor-yi Ng, Hiroshi Watabe.

**Visualization:** M. Rafiqul Islam.

**Writing – original draft:** M. Rafiqul Islam, Hiroshi Watabe.

**Writing – review & editing:** Mehrdad Shahmohammadi Beni, Chor-yi Ng, Mahabubur Rahman, Taiga Yamaya, Hiroshi Watabe.

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
