## [Decision Letter · Decision Letter 0]

22 Dec 2021

PONE-D-21-31024Proton range monitoring using 13N peak for proton therapy applicationsPLOS ONE

Dear Dr. Watabe,

Thank you for submitting your manuscript to PLOS ONE. After careful consideration, we feel that it has merit but does not fully meet PLOS ONE’s publication criteria as it currently stands. Therefore, we invite you to submit a revised version of the manuscript that addresses the points raised during the review process. When you resubmit your manuscript, please include a summary of the changes made and a succinct response to all recommendations or criticisms contained in the reports.

We look forward to receiving your revised manuscript.

Kind regards,

Mohammadreza Hadizadeh

Academic Editor

PLOS ONE

Journal Requirements:

This study was supported by Agency for Medical Research and Development (AMED) under Grant Number 20he2202004h0402.

[copy in statement]

Reviewers' comments:

Reviewer's Responses to Questions

**Comments to the Author**

1. Is the manuscript technically sound, and do the data support the conclusions?

Reviewer #1: No

Reviewer #2: Partly

Reviewer #3: Yes

2. Has the statistical analysis been performed appropriately and rigorously? 

Reviewer #1: N/A

Reviewer #2: Yes

Reviewer #3: Yes

3. Have the authors made all data underlying the findings in their manuscript fully available?

Reviewer #1: Yes

Reviewer #2: Yes

Reviewer #3: Yes

4. Is the manuscript presented in an intelligible fashion and written in standard English?

Reviewer #1: No

Reviewer #2: Yes

Reviewer #3: Yes

5. Review Comments to the Author

Reviewer #1: The authors describe a method of utilizing the auto-activation of nitrogen-13 positron emitters to monitor the location of Bragg peaks in proton therapy. The work builds upon a paper by Cho et al. published in 2017. The novel contributions of the work are to 1) develop a N-13 peak - Bragg peak offset vs proton energy function and 2) develop a method of spectral analysis to determine the N-13 component of a PET voxel signal.

With regard to contribution 1), I don’t believe a fitting function represents a significant contribution to the scientific community. Further, the precision of the data presented (whole integers) is on the same scale as the authors are attempting to fit to, which renders the fitting function obsolete.

With regard to contribution 2), the spectral analysis results appear promising, but no superior to the results presented by Cho et al. using an alternative method. Furthermore, as Cho et al. point out, N-13 peaks are only clearly present in monoenergetic Bragg peaks, which is of little practical use in proton therapy. Without performing the spectral analysis with spread-out Bragg peaks, the value of the work is questionable.

Considering the points above, I cannot recommend the work for publication. If the authors are able to demonstrate the spectral analysis method is superior to that of Cho et al. when applied to spread out Bragg peaks, the work should be resubmitted for consideration.

Reviewer #2: This is a very novel study by considering the generation of the positron emitters, especially the N13 peak proximal to the Bragg peak of proton therapy. The PHITS Monte Carlo simulation results show that the N13 peak is about 2 mm before the Bragg peak; therefore, the authors concluded that the N13 peak can be used to determine the Bragg peak location. However, I think it is still very challenging to apply the method in this study to the actual clinical treatment cases to monitor the proton ranges. My concerns are listed below:

(1) Can you elaborate the clinical significance of your study? For example, if it is a real patient being irradiated, is it practical to let the patient stay in the treatment couch waiting for 15 minutes to monitor the intensity of the positron emitters?

(2) The irradiated phantom is water-gel. I would say the reaction within the human tissues can be different from the water-gel. Have you considered the impact of the heterogeneities in human tissues on the N13 peak location?

(3) Only single energy beams have been considered in this study. Can you evaluate if this method is also applicable to using the SOBP, or even more complicated multi-field Intensity-modulated proton therapy cases?

(4) How will you accurately measure the spatial locations of these positron emitters? Using PET imagers around the patient during the proton therapy?

Reviewer #3: The paper contains interesting results in the reduction of the range uncertainties in proton therapy, and include useful tools developed that could help future research in this field, however, from my view, some parts of the work should be carefully reviewed before the publication.

General comments

In the Introduction to the work, various techniques used for the verification of the range in proton therapy collected in other works are summarized, however, references to previous papers from other authors based on the production of beta (-) emitters are not mentioned. From my perspective, these references have been used, but have not been properly referenced. In addition, it would be important to establish in a clear, explicit and summarized way, at the end of the Introduction, the main objective of the work and the methods followed.

As an example of other works with beta (-) emitters used but no mentioned in this paper would be: Simulation of Proton Therapy Treatment Verification via PET imaging of Induced Positron-Emitters, (2003), J.J. Beebe-Wang et al. The Figure 4 in your paper is exactly the same as Figure 3 of this paper.

In Materials and Methods part, however, the work does not have a continuity, and references are constantly made to other previous works by the same authors. In this way, to understand and follow the paper, it would be necessary to consult up to seven additional papers, as collected by the authors. It is understandable to refer to works and methodologies developed in previous works, but the paper should contain the information and data necessary to be fully understood without having to consult so many additional references. The paper could be organized as a compendium of some of your previous work. However, this is not the main goal stated by you.

As an example of this lack of continuity and permanent references to another works from the same authors would be:

Line 92 - More details regarding the MC simulation and modeling are available in our previous publications and the references therein [18-20].

Line 110 - the obtained results were normalized to the primary incident proton (see Ref [2]).

Line 135 - More details regarding compartmental modeling are available in our previous publications and our recently developed compartmental software [24-26].

Finally, in Conclusions, considering that in some of previous works, the use of 13N to reduce the uncertainty of the range in proton therapy have been already discussed, and it was concluded that although the signal from that isotope can be characterized by Monte Carlo techniques, however, there are difficulties in real measurements due to its low intensity, it would be important that the authors justify this fact in this work, and the real usefulness of this technique could be established in a more obvious way.

Some detailed comments

Line 3 – It is discovered. Sentence unclear. Did you discover that nuclear reaction? Please rewrite

Line 26 – Low exit dose. Sentence incorrect. Protons have not exit dose. Please rewrite

Line 63 – Recently, it was reported the reaction..Sentence incorrect. The reaction and the threshold energy is well known and stated long time ago. Please rewrite.

Line 98 - Table 1. Please justify why you choose this composition of the water-gel and why there is not Nitrogen in the phantom. In the same way, the percentage of Carbone is very high.

Line 107 - Table 2. Please include the correct reference where you have picked these data.

Line 144 - M is the number of radionuclides produced. Sentence unclear. From my view, M represents the different types of radionuclides produced, numbered from j=1 to j=M. Please rewrite.

Line 193 - From SRIM and Continuos Slow Down Approximation for protons it would be possible to obtain the results of Table 3. Please justify more widely how these data was obtained with PHITS.

Line 212 - My recommendation would be to include the average relative errors in the main paper, better than the caption of the Figure.

Line 237 - As already mentioned above, please include the correct reference.

Figure 1 - Please clarify in the paper that there is a gap of 25 cm of air from the proton beam to the phantom

In conclusion, considering that the results could be interesting in the field of range monitoring for proton treatment applications, my advice to the authors would be they should take the time to justify better and more extensively these results, and that the writing of the paper must contain references in a proper and kind way.

6. PLOS authors have the option to publish the peer review history of their article (what does this mean?). If published, this will include your full peer review and any attached files.

Reviewer #1: No

Reviewer #2: No

Reviewer #3: No

---

## [Author Response · Author response to Decision Letter 0]

13 Jan 2022

We appreciate the editor and all reviewers's comments, and we revised our manuscript accordingly. We believe the quality of the manuscript has been significantly by this updates. See the response letter for our responses.

---

## [Decision Letter · Decision Letter 1]

21 Jan 2022

Proton range monitoring using 13N peak for proton therapy applications

PONE-D-21-31024R1

Dear Dr. Watabe,

We’re pleased to inform you that your manuscript has been judged scientifically suitable for publication and will be formally accepted for publication once it meets all outstanding technical requirements.

Kind regards,

Mohammadreza Hadizadeh

Academic Editor

PLOS ONE

Additional Editor Comments (optional):

Reviewers' comments:

Reviewer's Responses to Questions

**Comments to the Author**

1. If the authors have adequately addressed your comments raised in a previous round of review and you feel that this manuscript is now acceptable for publication, you may indicate that here to bypass the “Comments to the Author” section, enter your conflict of interest statement in the “Confidential to Editor” section, and submit your "Accept" recommendation.

Reviewer #1: All comments have been addressed

Reviewer #2: All comments have been addressed

Reviewer #3: All comments have been addressed

2. Is the manuscript technically sound, and do the data support the conclusions?

Reviewer #1: Yes

Reviewer #2: Yes

Reviewer #3: Yes

3. Has the statistical analysis been performed appropriately and rigorously? 

Reviewer #1: N/A

Reviewer #2: Yes

Reviewer #3: (No Response)

4. Have the authors made all data underlying the findings in their manuscript fully available?

Reviewer #1: Yes

Reviewer #2: Yes

Reviewer #3: Yes

5. Is the manuscript presented in an intelligible fashion and written in standard English?

Reviewer #1: Yes

Reviewer #2: Yes

Reviewer #3: Yes

6. Review Comments to the Author

Reviewer #1: (No Response)

Reviewer #2: The authors have addressed all of my concerns. Thank you!

I would recommend the acceptance of its publication.

Reviewer #3: The points made in the initial review have been considered and corrected by the authors, so the paper is recommended for publication.

7. PLOS authors have the option to publish the peer review history of their article (what does this mean?). If published, this will include your full peer review and any attached files.

Reviewer #1: No

Reviewer #2: No

Reviewer #3: No

---

## [Editor Report · Acceptance letter]

7 Feb 2022

PONE-D-21-31024R1 

Proton range monitoring using 13N peak for proton therapy applications 

Dear Dr. Watabe:

I'm pleased to inform you that your manuscript has been deemed suitable for publication in PLOS ONE. Congratulations! Your manuscript is now with our production department. 

Kind regards, 

on behalf of

Dr. Mohammadreza Hadizadeh 

Academic Editor

PLOS ONE